# Management Practices of Bovine Mastitis and Milk Quality on Egyptian Dairies

**DOI:** 10.3390/vetsci10100629

**Published:** 2023-10-21

**Authors:** Heba S. Farag, Sharif S. Aly, Karima M. Fahim, Adel A. Fayed, Essam M. Abdelfattah, Samah M. El-Sayed, Yamen M. Hegazy, Wagdy R. ElAshmawy

**Affiliations:** 1Department of Internal Medicine and Infectious Diseases, Faculty of Veterinary Medicine, Cairo University, Giza 12211, Egypt; heba.saeed@cu.edu.eg (H.S.F.); adel.fayed@vet.cu.edu.eg (A.A.F.); samah.elsayed@cu.edu.eg (S.M.E.-S.); welashmawy@ucdavis.edu (W.R.E.); 2Department of Population Health and Reproduction, School of Veterinary Medicine, University of California, Davis, CA 95616, USA; 3Veterinary Medicine Teaching and Research Center, School of Veterinary Medicine, University of California Davis, Tulare, CA 93274, USA; eabdelfattah@ucdavis.edu; 4Department of Food Hygiene and Control, Faculty of Veterinary Medicine, Cairo University, Giza 12211, Egypt; dr.karima_fhc@cu.edu.eg; 5Department of Animal Hygiene, and Veterinary Management, Faculty of Veterinary Medicine, Benha University, Benha 13511, Egypt; 6Department of Animal Medicine (Infectious Diseases), Faculty of Veterinary Medicine, Kafrelsheikh University, Kafrelsheikh 33516, Egypt; yhegazy@kfu.edu.sa; 7Department of Clinical Studies, College of Veterinary Medicine, King Faisal University, Al-Ahsa 31982, Saudi Arabia

**Keywords:** Egyptian dairies, herd demographics, survey, management practices, mastitis status and control, milking practices

## Abstract

**Simple Summary:**

Dairy farms vary in their management practices, which may affect the incidence of mastitis and milk quality. The current study aims to describe different management practices in Egyptian dairies that may influence mastitis and milk quality. An in-person survey was conducted on 20 randomly selected Egyptian dairies. The questionnaire included questions about herd demographics, mastitis status and control, milking practices, and management practices of the study dairies. Most of the surveyed dairies had less than 500 lactating cows, with Holstein as the main cow breed. The average milk production in the study herds was 31.1 (SE ± 1.10) Kg/cow/day with open lots as the pen design for housing of lactating, dry, and close-up cows. Most of the study dairies milked their cows three times per day with herringbone as the most common parlor design. The study dairies used iodine as the most common disinfectant for both pre- and post-milking teat dip, with more than 50% of the dairies reporting that their milkers wore gloves during milking. The study dairies followed different milk quality management practices with 95% having reported inspection of udder health through visual inspection, 87% inspected udder hygiene score, and 72% inspected teat end scores. The diagnosis of subclinical mastitis was performed on 55% of the study dairies using CMT. Mastitis was reported at 52% of the milking herd, with 45.2% of the study herds having reported contagious mastitis pathogens. The main source for replacements was through importation of pregnant heifers.

**Abstract:**

Milk production continues to be the main source of income for dairy producers, and mastitis continues to be the major health challenge for dairy cows worldwide. The objective of the current study was to describe the different management practices implemented in Egyptian dairies, which may influence mastitis and improve milk quality. An in-person survey was completed with herd managers and owners of 20 Egyptian dairies selected using a stratified random sample from four of Egypt’s milk sheds. The questionnaire included 80 questions that inquired about herd demographics, mastitis status and control, milking practices, and management practices of the study dairies. Statistical analyses included descriptive statistics, multiple factor analysis (MFA), and hierarchical clustering to identify the important principal components and different dairy clusters. Of the surveyed herds, 69.50% had less than 500 lactating cows, with Holstein as the main cow breed. The reported average milk production on the study herds was 31.1 (SE ± 1.10) Kg/cow/day. Housing of lactating, dry, and close-up cows was in open lots. The majority of the dairies milked cows were three times per day (90.63%), the remaining milked cows four (5.00%) or a mix of four and three times per day. Furthermore, herringbone parlor design was the most common parlor design (66.79%) in the study dairies. The most common disinfectants used for both pre- and post-milking teat dip were iodine-based, 90.0% and 95.0%, respectively, while 52.16% of the dairies reported that their milkers wore gloves during milking. The reported mean annual percentage of mastitis was 52.3%, as a percent of the milking herd. The study dairies reported the inspection of udder health through either visual inspection (95.00%), udder hygiene score (86.88%), teat end score (71.88%), and/or using CMT (54.91%). Contagious mastitis pathogens were reported in 45.2% of the study herds. More than 50% of the study herds relied on importing pregnant cows or pregnant heifers as replacements. Multiple factor analysis identified 20 questions and represented 5 components of variability related to mastitis on dairies. The current survey of Egyptian dairies described the herd demographics and different management practices related to mastitis control and prevention. All the study dairies relied on the blanket intramammary antimicrobial drugs at dry off when current research elsewhere has identified selective dry cow therapy as an integral component of antimicrobial stewardship on dairies. Further research is required to identify the association between different management factors and the occurrence of mastitis.

## 1. Introduction

Milk production is the main source of income for dairy producers and mastitis continues to be the major health challenge for dairy cows worldwide. Previous studies reported high losses associated with mastitis on the dairies and estimated the cost of clinical mastitis per case to be EUR 428 [1], EUR 224 [2], and USD 735/lactation [3]. Dairy farms have implemented several preventive and control measures to minimize mastitis incidence, including proper milking practices, regular maintenance of milking machines, application of dry cow therapy, and culling of mastitis problem cows [3,4,5]. In addition, cow breed, herd size, facility design, bedding material, stocking density, presence of wash and holding pens, and the application of mastitis vaccines may influence mastitis incidence on dairies [6,7].

Prevalence of mastitis varies among countries and in between dairies due to the differences in environment and management practices. Environmental factors such as weather conditions, including temperature, humidity, and rain, affect the cleanliness of pens and, hence, cows, which may be associated with the prevalence of mastitis in a dairy primarily through the increase in bacterial counts on lactating cows [6]. Facility design, pen bedding, and its management influence the prevalence of mastitis in dairies as cows lay down with their udder and teats in contact with the bedding for 12–14 h per day [6,8]. Previous studies reported variability in the isolated mastitis pathogens with different facility designs [9]. Variations in the shade and bedding management between different facility designs may influence the incidence of mastitis; for instance, outbreaks of coliform mastitis are common in open lot dairies during the rainy seasons [6]. Bedding materials vary in their ability to provide a good medium for bacterial growth. Inorganic bedding materials such as sand have lower moisture and nutrient content, which makes them better than organic bedding (sawdust and recycled manure) in terms of being a risk factor for environmental mastitis [6]. Verbeke et al. (2014) reported a 1.5 times greater risk of mastitis due to any pathogen and a 2.57 times greater risk of mastitis due to E. coli in herds with cows that had dirty udders in comparison to herds with cows with clean udders [7]. A Chinese study reported an association between environmental mastitis pathogens isolated from clinical mastitis cases with varying bedding materials where Streptococcus dysgalactiae was more often isolated from herds with sand bedding, while Klebsiella spp. and other streptococci were more commonly isolated in herds with organic bedding [10]. The choice of bedding materials mainly depends on the availability of the bedding material and its cost, with producers preferring to use readily available inexpensive bedding [11]. Parlor hygiene and maintenance are very important in reducing the risk of mastitis, as cleaning and properly maintaining parlors will reduce gross contamination and slipping of the rubber cups during milking [12]. Similarly, teat dips play an important role in reducing clinical mastitis when applied properly. A previous study reported a reduction of 54% of intramammary (IMM) infections with major mastitis pathogens after the application of teat pre-dipping [13]. Proper application of the suitable post-milking teat dip is recommended to prevent the transmission of contagious mastitis pathogens such as *Mycoplasma bovis*, *Staphylococcus aureus* (*S. aureus*), and *Streptococcus agalactiae* (*S. agalactiae*) [12]. Proper milking hygiene includes wearing gloves, proper application of teat dips, removing the first 2–3 milk jets from each teat to screen for abnormal milk, and wiping dry the teats before attaching the milking machine [12,14]. Dairy breeds may vary in their susceptibility to mastitis. Washburn et al., (2002) reported a higher incidence of mastitis (41.2%) in Holstein cows in comparison to Jerseys (25.8%) [15], which may be attributed to differences in the innate immune response of the mammary gland as Jersey’s have higher milk somatic cell count in comparison to Holsteins, which may result in a lower prevalence of mastitis [16].

Understanding the influences of herd demographics, facility design, milking procedures, and various mastitis management practices on dairy farms is essential to reducing the impact of mastitis on dairy producers. To our knowledge there are no previously published studies on the influence of herd demographics, facility design, and milk quality management practices on mastitis and milk quality in Egyptian dairies. Hence, the objective of the current survey was to describe the management practices implemented on a random sample of 20 dairies with automatic parlors in Egypt’s four main milk sheds. The results of the current study will provide a description of the herd demographics and management practices that may influence milk quality in Egyptian dairies. In addition, the survey provides the baseline information for future studies on the risk factors associated with milk quality in Egyptian dairies.

## 2. Materials and Methods

### 2.1. Study Herds

The study was conducted between August 2021 and August 2022 using an in-person survey conducted with the herd manager or the owner of the dairy. The study was performed on a sample of 20 Egyptian dairy farms randomly selected from dairies in the four main milk sheds in the country. The study protocols were approved by University of California Davis’s IACUC (protocol number 21383) and Cairo University IACUC (protocol number VetCU10102019086). The selection criteria for study dairies included having an automated milking parlor and owner’s willingness to participate in the survey. The survey sampling frame was identified using a prior database generated from the Egyptian team members and a milk processor’s dairy list. Further, the database was updated using personal contacts of dairies in the four milk sheds. The four milk sheds were the Alexandria Desert Road and Beheira Governorate (n = 75 farms), the Ismailia Desert Road and Sharqia Governorate (n = 35 farms), the Faiyum and Beni Suef Governorates (n = 40 farms), and the Delta region (n = 50 farms). The four regions are referred from here onwards as Alexandria, Ismailia, Faiyum, and Delta. A stratified random sample with proportional allocation of dairies was selected from the updated database with regions as the strata resulting in a sample of 20 dairies out of the 200 dairies in the sampling frame (Table 1).

### 2.2. Survey Design

The survey questionnaire was drafted in English prior to being translated into the Arabic language. The questions were designed to identify the various milk quality management practices implemented on dairy farms in Egypt. The questionnaire was divided into the following four main sections: herd demographics, mastitis status and control, milking practices, and management practices, with a total of 80 questions.

Herd demographics. The first section included questions on the dairy location (region), the respondent’s role on the dairy, total number of cows, number of milking cows, average daily milk production (Kg), breed distribution within the herd, pen design for lactating, close-up and dry cows, average length of dry period, dry off protocol, and fresh cows (housing and milking). In addition, there were questions on which milk parameters were milk samples tested for [volume, somatic cell count (SCC), electrical conductivity (EC), and components of milk].

Mastitis status and control. The main focus of the second section was mastitis control. The questions addressed recording of mastitis cases, frequency of mastitis cases, on farm and laboratory mastitis diagnostics, mastitis treatment protocols and vaccination, use of internal or external teat sealants at dry off, and types of antimicrobial drugs (AMD) used at dry off. In addition, survey questions inquired about the use of diagnostic tests to assess udder health, such as the California mastitis test (CMT) and bacterial culture, udder hygiene, teat end scores, and visual inspection of abnormal milk. Participants were asked about differences in incidence of subclinical and clinical mastitis and culling due to mastitis across winter and summer seasons.

Milking practices. The third section focused on the different management practices adopted to reduce the occurrence of mastitis in the surveyed herds during milking. Questions discussed the number of milkings/cow/day, the layout of the parlor (parallel, parabone, herringbone, rotary, or other) and number of milking units. In addition, questions related to milkers inquired on the number of milkers, number of shifts/milker/day, and duration of a milkers shift. Management practices during milking such as pre-milking udder wash and drying, pre- and post-milking teat dips and type of teat dip disinfectant, wearing gloves during milking, the frequency gloves are changed, and type of towel used to dry the teat ends (cloth or disposable paper towel). Questions on cleaning and disinfection of the parlor and milking equipment included the frequency of cleaning and disinfection, type of the parlor cleaning system (automatic, manual), the used procedure (hot vs cold, acidic vs alkaline), and how the efficacy of parlor cleaning and disinfection are assessed. Finally, questions on parlor maintenance practices (replacing parts, intensive cleaning, adjusting vacuum pressure and cycle) were also included.

Dairy management practices. The fourth and last section addressed the general dairy management practices that may influence the occurrence of mastitis on the dairy such as pen bedding (material and frequency of changing the bedding), feeding (frequency, timing in regard to milking), and herd replacements (source, quarantine of new replacements, testing, and record keeping).

### 2.3. Statistical Analysis

Descriptive statistics. Frequencies as well as proportions and their standard errors (SE) were computed for categorical and ordinal variables, while means and their SE were computed for continuous variables. Confidence intervals for proportions were calculated using the normal approximation method. Data on the location of dairies in Egypt were classified into four regions Alexandria, Ismailia, Fayoum, and Delta based on the regional distribution. Herd size was categorized as <500 milking cows (small herds), 501–1500 (medium herds), and 1501–3500 (large herds). The variable bulk tank somatic cell count (BTSCC) (cells/mL) was classified into the following three categories: <200,000 cells/mL, 201,000–400,000 cells/mL, and >400,000 cells/mL [17]. To account for the survey design and produce estimates representative of the surveyed milk sheds [18], the Stata survey command “svyset” was employed to compute weighted proportions within the descriptive analyses and to compute survey adjusted standard errors and confidence intervals within the primary analyses. All descriptive statistics were performed using Stata 15 (Stata Corp, College Station, TX, USA).

Multiple factor analysis (MFA). MFA was conducted to identify the important principal components for the survey data [19]. A total of 73 variables were organized into 9 groups and included in the analysis as follows: herd demographics Group 1 (13 categorical variables) described the dairy’s location, herd’s breed distribution, pen design for lactating, close-up and dry cows, dry off protocol, and other variables related herd demographics; herd demographics Group 2 (4 continuous variables) described a herd’s number of lactating cows, average daily milk production, length of the dry off period, and average days in the fresh pen; mastitis diagnosis (10 categorical variables) described a dairy’s protocols for mastitis detection; mastitis treatment and preventive measures (11 categorical variables); mastitis status (2 continuous variables) described a herd’s incidence of clinical and subclinical mastitis; milking practices Group 1 (15 categorical variables) described a dairy’s milking practices and milking hygiene; milking practices Group 2 (8 continuous variables) described a dairy parlor’s number of milking units, number of milkers, number of shifts per milker, number of working hours per day, time elapsed from pre-dip and application of milking machine, milk storage temperature, and parlor vacuum pressure; management practices Group 1 (7 categorical variables) described bedding material used for cows at different production stages, where are cows fed after milking, source(s) of heifer replacements; and management practices Group 2 (3 continuous variables) described frequency of feeding of lactating cows and frequency of changing bedding materials during summer and winter. Principal components with correlation coefficients (coordinates) of 0.4 or greater were retained for interpretation [20]. Hierarchical clustering (HC) was performed on the MFA principal coordinates using the Ward’s criterion to aggregate individual dairies into relatively homogeneous subgroups (clusters). Both MFA and hierarchical clustering were performed in R software using the “FactoMineR” package [21].

## 3. Results

### 3.1. Descriptive Statistics

#### 3.1.1. Herd Demographics

The stratified random sample identified 7, 5, 4, and 4 dairies from Alexandria, Delta, Ismailia, and Faiyum regions, respectively (Table 1). Table 2 summarizes the results of the herd demographics section of the survey. The adult cow (lactating and dry cows) herd size of the surveyed herds ranged from 100 to 3468 cows, with 60.2% of the surveyed herds having less than 501 cows/herd, 20.1% had herd sizes from 501 to 1500, and 19.7% had more than 1500 cows. The number of lactating cows ranged from 80 to 1900 cows, with 69.5% of herds having less than 501 lactating cows, 25.1% ranging from 501 to 1500 cows, and only one herd (5.4%) having >1500 lactating cows. The cow breeds in the study dairies were Holstein, Brown Swiss, Montbeliarde, and Simmental. Holstein was the most common breed, with 65.5% of herds having ≥95% Holsteins and 34.5% had more than one breed. The mean daily milk production of the respondent dairies was 31.1 (SE± 1.10) Kg/cow/day, 59.2% of the dairies produced ≤ 32 Kg/cow/day, and the remaining 40.8% produced >32 kg/cow/day. The most common routinely tested milk parameter was milk volume (91.3%), followed by milk components (81.3%), and finally somatic cell counts (55.3%). Open lot was the only pen design for lactating, dry, and close-up cows on all the study dairies. The study dairies varied in their dry off schedule; specifically, 36.7% of the study herds dried off cows as needed, 32.2% weekly, and 21.2% monthly. More than 95% of the dairies had a separate pen for fresh cows, with 10.4% housed their fresh cows in the hospital pen for 1–2 days before moving them to a fresh pen, and only one dairy did not have a separate pen for fresh cows.

#### 3.1.2. Parlor Management and Milking Practices

A summary of the parlor management and milking practices of the study dairies is presented in Table 3. The average number of milking units per parlor was 25.3, with a minimum of 8 units/parlor and a maximum of 48 units/parlor. There were different parlor designs, with herringbone being the most common (66.8%), followed by parallel and parabone 17.5% and 15.7%, respectively. The median number of workers in the milking parlor per shift was 5 (Interquartile range (IQR= 4–8), and the median daily number of shifts per milker was 2 (IQR = 1–3) with a median of working hours of 3.25 (IQR = 2.5–6) completed by each milker per day. The majority of the dairies (90.6%) reported milking their cows 3 times daily, while 5.0% reported milking 4 times daily, and one dairy (4.4%) reported milking high-producing cows 4 times daily and low-producing cows three times daily. Regarding the milking practices, 90.0% of the dairies reported washing cow udders inside the milking parlor before milking; of those, 39.1% washed the udders for all cows, 24.8% washed only dirty udders, and 26.1% washed only dirty teats. All the study dairies reported stripping of the foremilk, while 95.0% reported the use of teat pre-dip, all reported the use of post-milking teat dip. Iodine was the most common disinfectant base used for both pre- and post-dip at 84.6% and 94.6%, respectively, and 45.1% of the dairies reported changing the disinfectant periodically. The most common application method for pre- and post-dip was via cup at 90.0% and 95.0%, respectively, followed by spray at 5.0% for both pre-dip and post-dip. The mean time elapsed between teat pre-dip and application of the milking machine was 66.3 (SE ± 9.58) seconds, with a minimum of 30 s and a maximum of 180 s. All the study dairies reported the use of disposable paper towels to dry the udder prior to milking, and approximately half of the dairies mentioned that their milkers wore gloves during milking, with 15.5% changing the gloves after each milking shift, 15.5% after each pen, 15.5% when the glove breaks, and 5.6% never changed their gloves. In 90% of the dairies, the milkers washed their hands either after each parlor turn (58.9%) or at the end of the milking shift (31.1%), while 10% did not report hand washing during the milking session. All dairies used a fixed vacuum pressure to milk all their cows with an average of 43.6 kPa (SE ± 0.52). All the study dairies reported maintaining their parlor equipment regularly, including activities such as intensive cleaning of the parlor equipment, adjusting the vacuum cycle and the vacuum pressure, and replacing the milk liners either every 2500–3000 (54.5%) or 3200–6000 (45.5%) milking times. The average temperature of the milk tank on the surveyed dairies was 2.9 °C (SE ± 0.12), with 52.9% of the dairies washing the milk tank after each milking and 47.1% washing the milk tank on a daily basis. Approximately 38.8% of the dairies reported bacterial cultures of swabs from the milk tanks and milk lines.

#### 3.1.3. Mastitis Management Practices

A summary of the different mastitis management practices in the study dairies is presented in Table 4. All the study dairies reported checking individual cow’s udder health through visual inspection for abnormal milk (95.0%), udder hygiene score (86.9%), teat end scores (71.9%), or conducting CMT (54.9%). The mean percentage of annual mastitis cases in the surveyed herds was 52.3% (SE ± 20.17) as a percentage of the milking cows in the herd. The surveyed herds reported an increase in the average mastitis cases with an increasing parity and during the winter season compared to summer. The highest frequency of mastitis was reported for fourth lactation cows, 12.3% during winter in comparison to 10.5% during summer season, while the lowest (5.0%) was reported for the first lactation cows during the summer season. Approximately 40.0% of the study dairies reported use of bacterial culture or polymerase chain reaction (PCR) for the diagnosis of mastitis, with 76.2% conducted as needed and 23.8% on all mastitis cases, while only one dairy used them for the diagnosis of all subclinical mastitis cases. The study dairies reported different protocols for the treatment of clinical mastitis, with all having reported the use of IMM AMDs and anti-inflammatory drugs. In addition, some dairies reported use of systemic AMDs (84.6%), fluid therapy (70.3%), oxytocin (45.8%), or increased milking times (42.9%). The commonly used IMM AMDs for the treatment of clinical mastitis were a combination of amoxicillin trihydrate and clavulanic acid (Synulox^®^, Zoetis UK Limited, Springfield Drive, UK), a combination of tetracycline, neomycin, bacitracin, (Mastijet Fort^®^, MSD Animal Health, NJ, USA), and cefquinome sulphate (Cobactan^®^ LC, MSD Animal Health, NJ, USA). More than 50% of the study dairies conducted antibiotic sensitivity testing to guide their mastitis treatment. Of those that conducted antibiotic sensitivity testing, 9.8% did so regularly on an annual basis and 90.2% conducted them only on cases that did not respond to treatment. Contagious mastitis was reported on 45.2% of the study dairies during the year prior to conducting the survey, specifically *S. aureus* (35.7%) and *S. agalactiae* (19.7%), while none of the surveyed herds reported *Mycoplasma bovis*. Dairies varied in their management of contagious mastitis cases, with 60.0% of the dairies reported treating infected cows, 13.3% culled infected cows, 13.3% practiced either treatment or culling depending on the health condition of infected cow, and 13.3% did not treat infected cows but isolated and milked them in a separate parlor.

The reported average percentages of subclinical mastitis on the surveyed herds were 15.9% and 17.4% during the summer and winter seasons, respectively. More than 90% of the surveyed herds diagnosed subclinical mastitis in their dairies, with CMT being the most common test (95.0%), followed by EC (39.5%) and somatic cell count (25.8%). Approximately 95% of the study dairies reported implementing a subclinical mastitis management protocol either through increasing milking times (83.8%) or using anti-inflammatories or IMM AMDs (16.2%). Of the surveyed dairies, 57.4% reported testing for SCC with 81.0% testing the bulk tank SCC and 19.0% testing both BTSCC and individual cow SCC through composite milk samples. The BTSCC of the study dairies ranged from 12,000 cell/mL to 550,000 cell/mL, with 69.6% reporting BTSCC < 200,000 cells/mL. The majority of the study dairies (94.6%) reported mastitis as a common reason to cull cows with a mean annual herd culling due to mastitis of 3.5%.

#### 3.1.4. Dry off Management Practices

All the study dairies changed the lactating cows’ total mixed ration prior to drying off to a low concentrate and high roughage ratio. A summary of the different management practices at dry off and during the dry period is summarized in Table 5. The average dry off period on the study dairies was 62.3 (SE ± 0.96) days, with 70.5% of the dairies reporting between 60 and 65 days. More than 80.0% of the study dairies practiced gradual cessation of milking to dry off their cows. The gradual cessation of milking for dry off was implemented through reducing the number of times cows were milked per day (for example, reducing milking times from three to two times per day for a week, followed by once-per-day milking during the following week before complete cessation of milking). All the surveyed dairies practiced blanket dry cow therapy at dry off with Cefalonium dihydrate (Cepravin^®^ DC, MSD Animal Health, NJ, USA), Ceftiofur Hydrochloride (Spectramast^®^ DC, Zoetis Inc., MI, USA), or benzathine cloxacillin (Orbenin^®^ DC, Merck Animal Health, NJ, USA). Only 9.37% of the study dairies reported the use of internal or external teat sealants at dry off; specifically, one dairy used an internal teat sealant, and another used an external teat sealant. Most of the respondent dairies believe that dry off treatment reduced the clinical mastitis rate in the subsequent lactation. The reported mean percentage of reduction in mastitis cases in the subsequent lactation due to the use of dry off IMM AMDs was 35.8%, and the reported mean increase in milk production was 9.83%.

#### 3.1.5. Farm Management Practices

A summary of farm management practices and sources of replacements is presented in Table 6. Our survey results showed that the most common bedding material used for lactating, dry, and close-up cows was sand (64.64%). The study dairies changed the bedding material 1.9 (SE ± 0.59) times and 6.2 (SE ± 1.98) times during the summer and winter seasons, respectively. Feeding times for lactating cows varied between dairies; the majority (60.2%) reported that they fed their lactating cows 3–5 times/day, 25.1% fed their cows 6–7 times/day, and 14.7% fed their cows 10–12 times/day. The main source for replacements on all the study dairies was home-raised heifers. In addition, 56.2% of the dairies reported buying replacements mainly through the importation of pregnant heifers (41.8%) or pregnant cows (14.4%). The majority of the study dairies (82.67%) quarantined newly purchased animals upon arrival and 6.68% cultured milk from newly purchased animals. Most of the study dairies (86.17%) kept records of newly purchased animal medications and approximately 85.0% tracked withdrawal periods for AMDs used to treat mastitis using computer software with or without paper records, and 15.0% used only paper records.

### 3.2. Multivariate Analysis

#### 3.2.1. Multiple Factor Analysis (MFA)

The first three dimensions of the MFA explained approximately 31.91% of the variability in the survey responses, 11.46%, 10.41%, and 10.03% for the first (Dim 1), second (Dim 2), and third (Dim 3) dimensions, respectively. The nine survey components modeled using the MFA and their correlation with the dimensions are depicted in Figure 1. The component Management Practices Group 1 contributed most of the variability explained in the second dimension, while Milking Practices Group 2 contributed most of the variability on the first dimension (Figure 1). Considering a 40% cut-off for contribution to the total variability in each dimension, we identified the three groups on Dim1 (Milking Practices 2, Milking Practice 1, and Herd Demographics 1), three groups on Dim 2 (Management Practice 1, Milking Practice 1, and Management Practice 2), and three groups on Dim 3 (Milking Practice 1, Management Practice 1, and Management Practice 2). Figure 2 depicts positively correlated components together and negatively correlated ones in the opposite quadrant.

The MFA analysis of 73 variables from the survey questions identified 20 questions that uniquely represented five components and had a correlation coefficient ≥0.4 on the first two dimensions (Table 7). Herd Demographics 1 and Milking Practices 2 accounted for 11.21% and 19.42% of the total variance on Dim 1, respectively, while Milking Practices 1 and Management Practices 1 and 2 accounted for 19.78%, 22.69%, and 14.79% of the total variability on Dim 2, respectively (Table 7).

#### 3.2.2. Hierarchical Clustering

Hierarchical clustering of survey responses partitioned the survey dairies into two clusters (Figure 3A,B); the profiles of each cluster are described in Table 8. Cluster 1 represented all four regions of the study dairies, while Cluster 2 included only dairies from Fayoum, Alexandria, and Delta regions, leaving out Ismailia. The majority of dairies in Cluster 2 reported that they dry off their cows as needed, while dairies in Cluster 1 mostly dry off their cows weekly or on a monthly basis. The majority of dairies in Cluster 1 reported that their milkers wore gloves during milking and changed gloves after every pen, while those in Cluster 2 did not wear gloves at the time of milking. More dairies in Cluster 1 indicated that they collected swabs from tanks and milk lines for bacteriological examination than dairies in Cluster 2. Dairies in Cluster 1 had more milking units in the parlor, more milkers, and working shifts than dairies in Cluster 2. The majority of dairies in Cluster 1 used sand as bedding material for dry, closeup, and lactating cows, while dairies in Cluster 2 used a combination of sand and soil as bedding material for cows. All dairies in Cluster 1 indicated that they changed bedding material less than 12 times/season during the winter and summer seasons, while dairies in Cluster 2 indicated that they changed bedding material more than 12 times during the winter season. The majority of dairies in Cluster 1 indicated replacing the liners of milking cups every 2000–3000 milking times, while the majority of dairies in Cluster 2 indicated that they changed the liners of milking cups every 3200–6000 milking times.

## 4. Discussion

The current study is the first descriptive study based on a random sample of the dairies representing the four main milk sheds in Egypt and focused on the management practices that influence milk quality and mastitis in Egyptian dairies. Understanding the differences in herd characteristics and management practices in Egyptian dairies can provide guidance for the industry to improve their practices, leading to increased profitability and sustainability.

### 4.1. Herd Demographics

Most of the surveyed herds (69.5%) were small sized herds with less than 500 lactating cows (range from 80 to 1900 lactating cows), with Holstein as the main breed and an average milk production of 31.1 kg/day. Most of the study dairies’ producers tested their lactating herds routinely for milk volume and components, while only 50% tested for SCC. The main pen design for lactating, dry, and close-up cows was an open lot, and the majority of dairies had a separate fresh pen. The herd demographics was an identified component variable with high variability on the first dimension of the factor analysis. In addition, the hierarchical cluster analysis revealed that large dairies, otherwise more prevalent in the Alexandria Desert Road and Beheira Governorate, were mostly represented in Cluster 1. In comparison to Cluster 2, which was predominantly dairies in the Delta region, represented smaller herds. Vimbalkar et al., 2020 reported that herd size, landholding, and dairy income were positively associated with the probability of implementing subclinical mastitis diagnosis in dairies in India [22]. Jayarao et al., 2004 reported that herd size was associated with different farm management practices, such as the type of milking facility, times cows are milked per day, use of automatic milking machine removal, type of bedding, and mastitis control practices [23]. Krattley-Roodenburg et al., 2021 found an association between the herd size and the timing of cow milking after calving, as larger herd size farms milk all their cows within 2 h after calving [24]. Shock et al., 2015 reported an association between the herd size and BMTSCC during the summer season in Ontario, Canada, as small herds reported an increase in the BMTSCC during the summer season in comparison to larger herds due to variations in the management practices [25]. Dong et al., 2012 also reported that large farms had a lower SCC limit in comparison to small herds [26].

### 4.2. Parlor Management and Milking Practices

Most of the study dairies milked their cows three times daily, with herringbone parlor being the most common design and a median number of workers/parlor/shift of 5 working 3.25 h/milker/day. Pre- and post-milking udder hygiene, including udder wash, pre-dip, stripping of the foremilk, wiping of teats, the time between pre-dip and application the of milking machine, and post-dip will impact the quality of milk produced on the dairy [27,28,29,30]. The current survey assessed the milk quality practices of the study dairies including udder washing inside the milking parlor (90.0%), with 39.1% washing the udder for all cows, 24.8% washing only dirty udders, and 26.1% washing only dirty teats. In addition, the study dairies practiced stripping of the foremilk, using post-milking teat dips, and using paper towels to dry the teats on all of the study dairies, only 95.0% of the farms used teat pre-dip, and about 50% of the dairies reported that milkers wore gloves during milking. Manual foremilk stripping is beneficial to dairies as it helps in the visual identification of cows with clinical mastitis, removing the foremilk, which contains the highest bacterial count, stimulating oxytocin release and milk ejection, and reduce teat damage as cows exposed to manual forestripping showed lower odds (0.3) of teat tissue changes and higher milk ejection capacity in comparison to controls [31]. The most common disinfectant used for both pre- and post-milking teat dip was iodine-based and applied using a cup (90.0% and 95.0% for pre-dip and post-dip, respectively). The MFA analysis showed that the pre-dip methods and pre-dip disinfectant base were highly correlated with the variability in the dataset with the use of a cup to apply an iodine-based pre-dip as the most common for both clusters while the spray method, chlorohexidine, and lactic acid were only used in Cluster 1. Dufour et al., 2011, found that the use of gloves during milking, application of post-milking teat dips, and regular maintenance of the milking system was associated with lower SCC [27]. In addition, free-stall housing, use of sand bedding, and application of blanket dry-cow therapy were also associated with lower SCC [27]. Enger et al., 2015 concluded differences in the sensitivity of different mastitis pathogens to different teat dips and contact times of 30 s and 15 s may be optimal for iodophors and H_2_O_2_ pre-milking teat dips, respectively [28]. Quirk et al., 2012 reported a reduction in IMM infections for quarters with their teats dipped with iodine post-milking in comparison to non-dipped quarters [29]. Boddie et al., 2004 reported a significant reduction in *S. aureus* and *S. agalactiae* following the use of 0.1% iodine post-dip [30]. Pankey, 1989 mentioned that pre-milking udder sanitation, including water hose cleaning, using wet towels for cleaning and dry towels to dry the teats, and the application of teat disinfectants was associated with reduced microbial load on the teats and reduced the environmental mastitis [14]. The mean pre-milking lag time for the study dairies was 66.3 s with a minimum of 30 s and a maximum of 180 s, which falls within the recommended pre-dip contact time and oxytocin release to avoid bimodal milking, reduced milking efficiency [32,33]. Watters et al., 2012 reported the highest incidence of bimodal milking when the lag time was either zero or 240 s [33]. Regular maintenance of milking equipment is essential for cow health, welfare, and productivity. All the study dairies reported regular intensive cleaning of the parlor equipment, adjusting the vacuum cycle and the vacuum pressure in addition to replacing the milk liners either every 2500–3000 (54.5%) or 3200–6000 (45.5%) milking times. The factor analysis revealed that the frequency of replacing the liners was correlated with the variability in the data, with more than 60% of Cluster 1 dairies replacing the liners every 2000–3000 milking times, while 66.67% of dairies in Cluster 2 replaced the liners every 3200–6000 milking times. The wide variation in the changing of the milk liners could be attributed to the variation in the number of milking cows in the study herds (ranging from 80 to 1900 cows). In addition to parlor maintenance, the study dairies reported washing the milk tank after each milking or at least daily; more than half of the dairies checked the efficacy of the disinfectants, and about one-third cultured the milk tank and milk lines for bacteria.

### 4.3. Mastitis Management Practices

The regular monitoring of udder health and surveillance for mastitis are important tools for the prevention and control of mastitis on dairies. All of the study dairies reported inspection of udder health through visual examination, udder hygiene score, teat end score, and/or using CMT. The mean annual percentage of mastitis cases was 52.3% out of the milking cows, and the average of subclinical mastitis was 17.4% during winter and 15.9% during summer seasons. The prevalence of mastitis on dairies varied at the international, national, and farm levels due to seasonal variations, farm management, and cow-related factors. Girma et al., 2022 reported the overall prevalence of mastitis in Ethiopia as 43.6%, with 12.6% for clinical mastitis and 32.2% for subclinical mastitis, respectively [34]. Abed et al., 2021 reported a prevalence of 46.0% for subclinical mastitis use CMT in two Egyptian herds [35]. Moosavi et al., reported a mean incidence rate of 30% for clinical mastitis in a study in Iran between 2005 and 2008 [36]. Ramirez et al., 2014 reported a mean prevalence of 37.2% for subclinical mastitis in Colombia [4]. In addition to the farm management factors, mastitis is influenced by cow-related factors, including breed, parity, productivity, stage of lactation, and previous mastitis history [4,15]. Aly et al., 2022 reported a significantly lower clinical mastitis during the first 150 DIM for Holstein Jersey cross breeds and Jerseys in comparison to Holstein breeds [37]. Washburn et al., 2002 reported the doubling of clinical mastitis cases per cow for Holsteins in comparison to Jerseys [15]. Aly et al., reported that cows with ≥ third lactation had 1.7 higher odds of developing clinical mastitis during the first 150 DIM in comparison to second lactation cows [37]. Previous studies reported a seasonal variation in the mastitis cases among different farms with high prevalence during the wet season (fall/winter) in comparison to dry season (spring/summer) [36,37,38].

The study dairies regularly used different tools to check for udder health, such as visual inspection (95.0%), udder hygiene score (86.6%), teat end scores (71.8%), and CMT (54.9%). Visual inspection continues to be a common method for identifying clinical mastitis in dairies and improving cow and udder hygiene will reduce mastitis, and improve milk quality in dairies [4,39,40]. Aly et al., 2022 found that cows with higher teat end scores at dry off (two or more teats with a teat end score of 4) had 2.7 times higher odds for mastitis within 150 days post-calving in comparison to cows with lower teat end score (one or none of the teats had teat end scores of 4) [37]. Verbeke et al., 2014 reported that dairy herds with dirty udders had a higher incidence rate for clinical mastitis in comparison to herds with clean udders (9.0 vs 6.0 quarter cases per 10,000 cow days) [7]. Early identification of subclinical mastitis using CMT, SCC, and EC will reduce clinical mastitis cases and improve dairy profitability. California mastitis test continues to be the main tool for the identification of subclinical mastitis in Egyptian dairy herds, greater than 90% of the study dairies used CMT, while 34.5% relied on EC, and 25.8% used SCC to identify subclinical mastitis cows. Sargeant et al., 2001 reported that CMT is a useful tool for the identification of fresh cows with IMM infections [41]. In addition, Claycomb et al., 2009 reported that automated inline EC can be used as a useful tool for the early detection of mastitis in dairies [42]. The most common practice for the management of subclinical mastitis on the study dairies was increasing the milking times per day (83.8%), followed by using IMM AMDs and anti-inflammatories (16.2%). Kromker et al., 2010 found no significant difference in clinical and microbiological cure of clinical mastitis between cows who received IMM AMDs alone or a combination of IMM AMDs and increased frequency of milking [43]. Waterman et al., 1983 reported no significant difference in udder health, IMM infections, and SCC between cows milked three times and cows milked twice daily [44].

Contagious mastitis was reported by 45.2% of the study herds, with *S. aureus* being the most common, followed by *S. agalactiae,* and none of the study herds reported *Mycoplasma bovis*. The main management practice of contagious mastitis in the study dairies was through treatment (60 % of the infected herds) followed by culling or milking in a separate pen. Despite culling being the recommended practice to control contagious mastitis, Egyptian herds treat such infections, which may be attributed to the small herd size or the higher prevalence as a previous study reported the isolation of *S. aureus* and *S. agalactiae* (44.9% and 22.1%, respectively) from milk samples of subclinical mastitis cases on two Egyptian dairy herds [35].

Blanket antimicrobial treatment continued to be the major treatment protocol for clinical mastitis. All of the study herds used IMM AMDs and anti-inflammatories for the treatment of clinical mastitis and some reported the use of systemic antimicrobials, fluid therapy, oxytocin, and increase milking times per day in their treatment protocols. Blanket IMM AMDs treatment for all mastitis cows is not the ideal protocol for treatment as bacteriological culture of clinical mastitis samples revealed that about 30% of the samples had no growth [45,46], which means that the use of IMM AMDs in about 30% of clinical mastitis is not necessary and increases the risk for AMR. More than 50% of the study dairies reported conducting antibiotic sensitivity as needed (90.2%) or regularly (9.8%) to test for antimicrobial resistance against the commonly used IMM AMDs and guide treatment protocols. Antibiotic sensitivity is a tool to monitor the efficacy and optimize mastitis AMD treatment protocols [47].

### 4.4. Dry off Management Practices

The management practices of dairy cows at dry off are important due to their impact on productivity in the following lactation. The average dry off period for the current study dairies was 62.3 days, which matches the recommended dry period length to maximize milk production in the subsequent lactation [48,49]. A dry off period of 60 days is required to allow for the regeneration of the mammary gland, cure current IMM infections, and restore the cow’s body reserves all of which impact the following lactation’s productivity. Gott et al., 2017 reported the reduction in milk production during the subsequent lactation for cows with short dry off periods [49]. Newman et al., 2009 reported that more than 80% of dry off IMM infections were cured during the dry period [48]. Most of the study dairies practised gradual cessation of milking as a dry off strategy by reducing the milking times and changing the ration composition to have more fiber and less concentrates. A previous study reported no difference in the milk production and SCC in the following lactation between different milk cessation methods (abrupt vs gradual) at dry off [49]. All study dairies used blanket dry cow therapy through the infusion of IMM AMDs into each quarter at dry off after the last milking and only 9.4% used external or internal teat sealants. Due to the growing concern of antimicrobial resistance worldwide recent studies aimed at judicious use of AMDs on dairies and implemented the use of selective dry cow therapy to reduce the use of antibiotics on dairies [37,50].

### 4.5. Farm Management Practices

Farm management practices, including pen bedding material, bedding change frequency, feeding frequency, and sources of replacements, may influence the prevalence of mastitis and milk quality in dairies [11,51]. The most common bedding material used on the study dairies was sand for lactating, dry, and close-up cows, while the feeding frequency for lactating cows varied between dairies, with three times/day being the minimum and a maximum of 12 times/day. Pen bedding material and frequency of changing bedding were correlated according to the MFA, with sand being the most common bedding material in Cluster 1 dairies and soil as the most common in Cluster 2 dairies. The type of bedding materials is an important risk factor for environmental mastitis, as cow teats may be in contact with bedding material for approximately 12 h daily. Organic bedding material (dried manure, straw, sawdust, and wood shavings) may provide a suitable environment for pathogens to grow and multiply in contrast to the inorganic bedding (sand, washed sand) [11].

Replacements on dairy farms may constitute a risk factor for the spread of infectious diseases with a special emphasis on contagious mastitis so it is important to select the source of replacements, apply quarantine measures, and test for contagious mastitis before introducing them into the main herd. More than 50% of the study dairies reported buying replacements through the importation of pregnant heifers and pregnant cows with about 80% having applied quarantine measures and only 6.7% reported the culture of milk samples of replacements. Previous studies reported that contagious mastitis frequency was higher in dairy herds that purchased replacements in comparison to closed herds [51,52].

## 5. Limitations

The current study is the first descriptive study on the herd demographics, milking practices, parlor management, mastitis management, dry off protocols, and farm management practices that may influence milk quality and mastitis occurrence based on a random sample of Egyptian dairies representing the country’s four main milk sheds. Longitudinal studies are required to investigate the associations identified using the current survey between management practices and the incidence of mastitis on the Egyptian dairies. The current survey reported that all the study herds implemented the blanket IMM AMDs for the treatment of clinical mastitis and at dry off; however, further research is required to quantify the impact of such a practice on development of AMR. In addition, future clinical trials are required to evaluate the impact of application of selective IMM AMD treatment of clinical mastitis cases and selective dry cow therapy as stewardship tools for judicious use of AMD on Egyptian dairies.

## 6. Conclusions

The majority of Egyptian dairy herds have less than 500 lactating cows, with Holsteins being the predominant breed milked three times per day. The surveyed herds clustered into two main groups based on their location and management practices, with larger herds predominantly in Cluster 1. Compared to Cluster 2, Cluster 1 cows were more frequently dried off, prepared prior to and after milking with more variable teat dip formulations, and milked in parlors where milking machine liners were more frequently replaced. The surveyed dairies implemented different management practices to reduce mastitis, such as washing the udder before milking, visual inspection of the udder and milk before attaching the milking machine, and using pre- and post-milking teat dips. Iodine-based disinfectants were commonly used for both pre- and post-milking teat dips. Contagious mastitis pathogens were reported on 45.2% of the study herds, with *S. aureus* and *S. agalactiae* being the most common. All the study dairies relied on raising their own heifers; in addition, more than 50% relied on the importation of either pregnant cows or pregnant heifers as a source for replacements, with only 7.8% routinely culturing the milk samples from new replacements. All the study dairies implemented blanket IMM AMD treatment for clinical mastitis cases and at dry off. Research on the development and application of selective dry cow therapy on Egyptian dairies to reduce antibiotic use and AMR is required.

## Figures and Tables

**Figure 1 vetsci-10-00629-f001:**
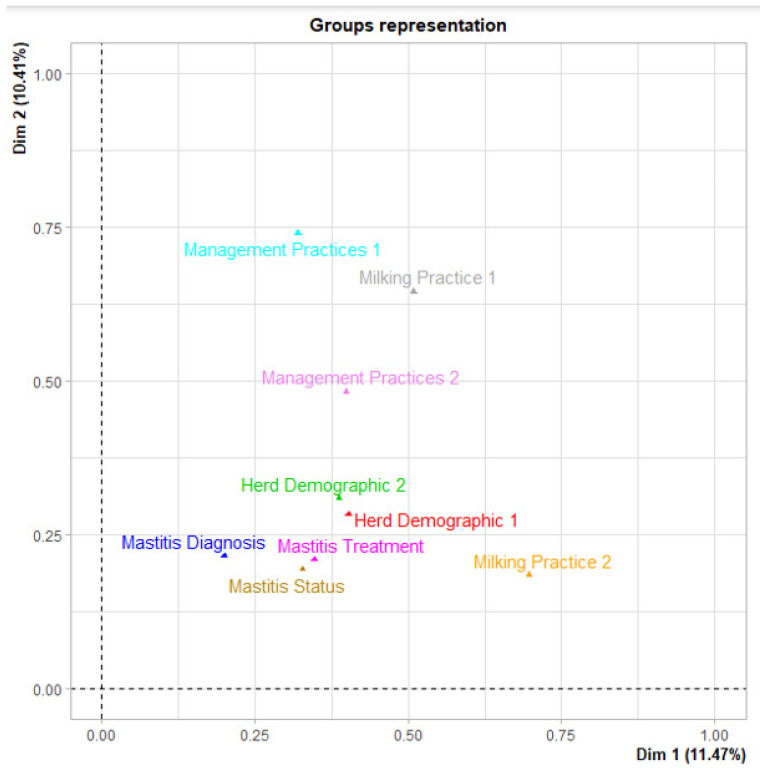
Representation of groups (components) on the first and second dimensions of the multiple factor analyses of 73 survey variables from a stratified random survey of Egyptian dairies to collect information on management and milking practices related to mastitis in cattle. Three groups (Herd Demographics 1, Milking Practices 1, and Milking Practices 2) have the largest contribution to the first dimension. In the second dimension, the three groups (Milking Practices 1, Management Practices 1, and Management Practices 2) have the highest contribution to the second dimension. The named components are identified within dimension by unique colors.

**Figure 2 vetsci-10-00629-f002:**
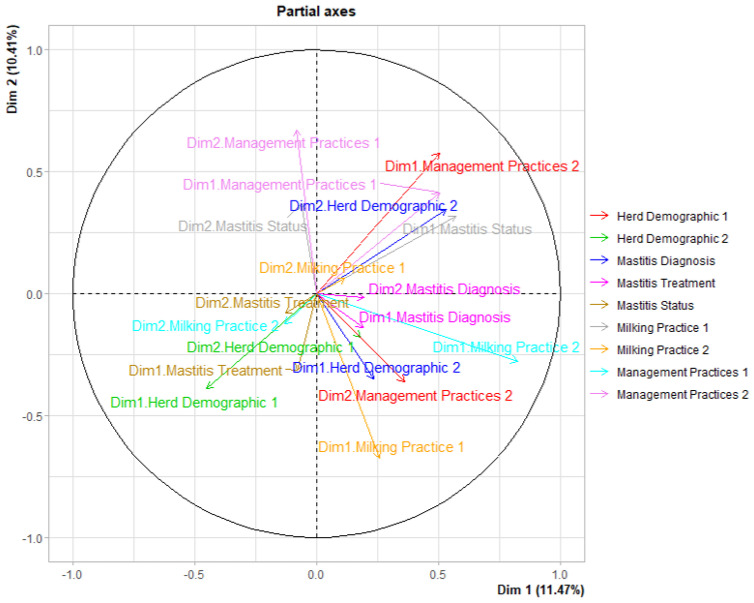
Illustration of the relationship between the MFA components and correlation between components and dimensions. Components that are positively correlated are depicted together and those negatively correlated are in the opposite quadrants. The groups with correlation coefficients (coordinates) of 0.4 or greater were retained for interpretation.

**Figure 3 vetsci-10-00629-f003:**
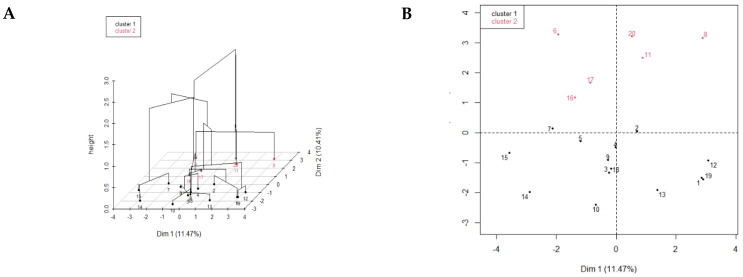
(**A**,**B**) Representation of the two clusters identified using the results of the multiple factor and hierarchical clustering analyses of the survey responses obtained from 20 dairies in Egypt to collect information on management and milking practices related to mastitis in cattle.

**Table 1 vetsci-10-00629-t001:** Summary of the distribution of the study dairies between the four milk sheds in Egypt.

District	Number of Dairies	Represented % of Dairies	Number of Study Dairies
Alexandria Desert Road and Beheira Governorate	75	37.5%	7
Ismailia Desert Road and Sharqiyah Governorate	35	17.5%	4
Fayoum and Beni Suef Governorates	40	20.0%	4
Delta region	50	25.0%	5
Total	200	100.0%	20

**Table 2 vetsci-10-00629-t002:** Herd demographics of a random sample of Egyptian dairy farms participated in the milk quality and mastitis management practices survey (n = 20).

Herd Demographics	Levels	Number of Farms	Estimate (SE)	95% CI	
				Lower	Upper
Respondent role	Owner of farm	6	29.72 (7.95)	15.81	48.77
	Veterinarian	3	16.98 (7.59)	6.09	39.20
	Owner and veterinarian	2	10.94 (7.74)	2.21	40.06
	Others ^1^	8	42.36 (10.03)	23.44	63.81
Herd size (total number of cows/herd)	0–500	12	60.18 (12.06)	34.21	81.45
	501–1500	4	20.09 (9.59)	6.62	47.15
	1501–3500	4	19.73 (9.89)	6.14	48.02
Number of lactating cows/herd	0–500	14	69.55 (11.25)	42.56	87.57
	501–1500	5	25.09 (10.82)	9.00	53.15
	1501–3500	1	5.36 (5.36)	0.60	34.71
Average daily milk production/cow	18–32 kg	12	59.20 (11.79)	34.01	80.33
	>32 kg	8	40.80 (11.79)	19.67	65.99
Breed ^2^	Pure Holstein	13	65.54 (11.66)	38.91	85.03
	Mixed herd	7	34.46 (11.66)	14.97	61.09
Milk parameters routinely tested	Milk volume	18	91.25 (5.05)	73.17	97.55
	Somatic cell count	11	55.18 (12.10)	30.45	77.58
	Milk components	16	81.25 (7.94)	58.94	92.90
	Electrical conductivity	9	44.46 (11.66)	22.74	68.53
Dry off schedule	Weekly	6	32.17 (11.61)	13.24	59.59
	Every 2 weeks	1	5.28 (5.28)	0.58	34.61
	Every 3 weeks	1	4.62 (4.62)	0.52	31.17
	Monthly	4	21.23 (10.03)	6.98	49.17
	As needed	7	36.70 (11.51)	16.78	62.50
Housing of fresh cows	Separate fresh cow pen	16	78.93 (9.08)	54.08	92.26
	Separate fresh cow pen and hospital pen	3	15.71 (8.53)	4.54	42.22
	Regular lactating pen	1	5.36 (5.36)	0.60	34.71
Duration in fresh pen	Up to 21 days	7	35.75 (10.70)	17.09	60.03
	22–40 days	8	42.36 (11.93)	20.59	67.56
	41–83 days	4	21.89 (10.35)	7.16	50.45

^1^ Others = executive director, nutrition officer, and parlor official. ^2^ Pure breed if the herd has >=95% of cows of the same breed; mixed breed, if the herd contains more than one breed including Holstein, Brown Swiss, Simmental, and Montbéliard.

**Table 3 vetsci-10-00629-t003:** Summary of responses to the parlor management and milking practices of a random sample of Egyptian dairy farms participated in the milk quality and mastitis management practices survey (n = 20).

Parameter	Levels	N	Estimate (%)	95% CI	
				Lower	Upper
Milking times per day	Three times	18	90.63 (6.64)	64.82	98.07
	Four times	1	5.00 (5.00)	0.56	32.89
	Three and four times	1	4.37 (4.37)	0.50	29.57
Parlor design	Herringbone	13	66.79 (8.53)	47.08	81.97
	Parallel	4	17.50 (0.00)	17.50	17.50
	Parabone	3	15.71 (8.53)	4.54	42.22
Average number of milking units per parlor		20	25.3 (4.02)	16.88	33.72
Udder wash	Yes	18	90.00 (6.12)	68.02	97.44
	No	2	10.00 (6.12)	2.56	31.98
	All udders	8	39.11 (8.90)	22.53	58.65
	Only dirty udders	5	24.82 (9.11)	10.50	48.16
	Only teats	5	26.07 (9.74)	10.78	50.73
	Inside the milking parlor	18	90.00 (6.12)	68.02	97.44
Pre-dip disinfectant base	Iodine	17	84.64 (8.14)	59.39	95.41
	Iodine + chlorohexidine	1	5.36 (5.36)	0.60	34.71
	Lactic acid	1	5.00 (5.00)	0.56	32.89
Pre-dip method	Cup	18	90.00 (6.12)	68.02	97.44
	Spray	1	5.00 (5.00)	0.56	32.89
Post-dip disinfectant base	Iodine	19	94.64 (5.36)	65.29	99.40
	Iodine + chlorohexidine	1	5.36 (5.36)	0.60	34.71
Post-dip method	Cup	19	95.00 (5.00)	67.11	99.44
	Spray	1	5.00 (5.00)	0.56	32.89
Average time elapsed between pre-dip and application of milking machine	(sec.)	20	66.25 (9.58)	46.1	86.3
Do you periodically change disinfectants?	Yes	9	45.09 ± 11.38	23.66	68.51
	No	11	54.91 ± 11.38	31.49	76.34
Do milkers wear gloves during milking?	Yes	10	52.16 (12.22)	27.73	75.60
	No	9	47.84 (12.22)	24.40	72.27
How often milkers change their gloves?	After each milking	3	15.51 (8.98)	4.08	44.18
	After every pen	3	15.51 (8.98)	4.08	44.18
	When take a break	3	15.51 (8.98)	4.08	44.18
	Never change	1	5.64 (5.64)	0.62	36.39
How often milkers wash their hands during milk shifts?	After each cycle	12	58.93 (10.95)	35.48	78.92
	At the end of each period	6	31.07 (10.95)	13.23	57.13
	No hand washing	2	10.00 (5.77)	2.77	30.21
Type of towels used to dry the udder	Papers (disposable wipes)	20	100	.	.
Average storage temperature of milk in bulk tanks (◦C)		20	2.9 (0.12)	2.65235	3.14765
How regularly are milk tanks washed?	After each milking	11	52.86 (7.33)	37.54	67.65
	Daily	9	47.14 (7.33)	32.35	62.46
Do you check the efficiency of the tank disinfectant?	Yes	11	56.89 (10.03)	35.57	75.93
	No	8	43.11 (10.03)	24.07	64.43
Do you culture swabs from tanks and milk lines?	Yes	8	38.84 (11.38)	18.70	63.68
	No	12	61.16 (11.38)	36.32	81.30
Describe your parlor maintenance	*Frequency of replacing liners of milking cups:*				
	Every 2500–3000 milking times	11	54.55 (12.14)	29.83	77.22
	Every 3200–6000 milking times	9	45.45 (12.14)	22.78	70.17
	*Intensive cleaning*	20	100	.	.
	*Adjusting vacuum cycle and pressure*	19	100	.	.

**Table 4 vetsci-10-00629-t004:** Summary of the responses about mastitis status and control of a random sample of Egyptian dairy farms participated in the milk quality and mastitis management practices survey (n = 20).

Parameter	Levels	Number of Farms	Estimate (SE)	95% CI	
				Lower	Upper
Farmers check udder health of individual cows	Check the udder hygiene score	17	86.88 (4.37)	74.58	93.72
	Check the teat end score	14	71.88 (9.04)	49.77	86.83
	Conduct CMT ^1^	11	54.91 (11.39)	31.49	76.34
	Visual inspection for abnormal milk	19	95.00 (5.00)	67.11	99.44
Diagnosis of mastitis by bacterial culture or PCR ^2^	Yes	8	40.80 (11.79)	19.67	65.99
	SCM ^3^ (every time)	1	4.37 (4.37)	0.50	29.57
	CM ^4^ (as needed)	6	76.15 (15.92)	28.67	96.21
	CM (every time)	2	23.85 (15.92)	3.79	71.33
Annual mastitis cases	(% of herd)	20	52.34 (20.17)	9.95	94.73
Percentage of mastitis recorded during summer and winter seasons for each parity	First lactation–summer	4	5 (3.34)	−5.63	15.63
	First lactation–winter	4	6.25 (2.98)	−3.24	15.74
	Second lactation–summer	4	7.25 (3.61)	−4.25	18.75
	Second lactation–winter	4	9 (3.08)	−0.81	18.81
	Third lactation–summer	4	7.75 (2.17)	0.83	14.67
	Third lactation–winter	4	9.25 (1.88)	3.24	15.25
	Fourth lactation–summer	4	10.5 (4.55)	−3.99	24.99
	Fourth lactation–winter	4	12.25 (4.09)	−0.76	25.26
Clinical mastitis treatment protocol	Use of IMM AMD ^5^	20	100		
	Use of anti-inflammatory	20	100		
	Use of systemic AMD ^6^	17	84.64 (7.88)	60.41	95.22
	Use of fluid therapy	14	70.27 (9.89)	46.42	86.57
	Increase milking times	9	42.86 (5.36)	32.05	54.39
	Use of oxytocin	9	45.80 (10.08)	26.33	66.65
	Others ^7^	3	14.73 (8.53)	3.93	42.18
Common IMM AMD used to treat clinical mastitis	Synulox^®^	12	60.54 (11.66)	35.29	81.19
	Mastijet Fort^®^	11	54.20 (11.25)	31.17	75.56
	Cobactan^®^ LC	11	55.18 (12.06)	30.45	77.58
	Tetra-Delta^®^	3	14.73 (8.53)	3.93	42.18
	Multiject^®^	3	15.36 (7.88)	4.78	39.59
	Spectramast^®^ LC	3	15.36 (7.88)	4.78	39.59
	Mamifort^®^	2	10.90 (7.71)	2.20	39.93
	Albadry Plus^®^	1	4.37 (4.37)	0.50	29.57
Conduct antibiotic sensitivity to guide mastitis treatment		11	54.82 (11.18)	30.50	77.04
Antibiotic sensitivity protocol	Regularly	1	9.77 (9.77)	0.78	59.82
	For special cases	10	90.23 (9.77)	40.18	99.22
Herds reported contagious mastitis cases during last year		9	45.18 (11.88)	22.96	69.50
Contagious mastitis pathogens	*Staphylococcus aureus*	7	35.80 (11.25)	16.50	61.15
	*Streptococcus agalactiae*	4	19.73 (9.89)	6.14	48.02
	*Mycoplasma bovis*	0	0		
Contagious mastitis cases management	Treatment	5	60.00 (18.76)	19.11	90.50
	Culling	1	13.33 (13.21)	1.02	69.67
	Treatment or culling	1	13.33 (13.21)	1.02	69.67
	No treatment and milk separately	1	13.33 (13.21)	1.02	69.67
Diagnosis of SCM	California Mastitis Test	19	95.00 (5.00)	67.11	99.44
	Electric conductivity	8	39.46 (12.01)	18.34	65.42
	Somatic cell count	5	25.80 (8.75)	11.66	47.81
	Others ^8^	1	5.00 (5.00)	0.56	32.89
Average percentage of subclinical mastitis	Summer	12	15.93 (7.16)	0.17	31.69
	Winter	12	17.42 (5.77)	4.69	30.13
Treatment protocol for SCM		19	94.64 (5.36)	65.29	99.40
Subclinical mastitis treatment protocol	Increase milking times	16	83.77 (9.37)	54.28	95.74
	Others (anti-inflammatory, or IMM AMD)	3	16.23 (9.37)	4.26	45.72
Test for somatic cell count (SCC)		12	57.42 (11.84)	32.45	79.11
Test for somatic cell count (SCC)	Bulk tank	10	81.01 (12.62)	40.66	96.37
	Bulk tank and composite sample from all 4 quarters	2	18.99 (12.62)	3.63	59.34
Bulk tank somatic cell count	<200,000 cells/ml	8	69.57 (13.56)	35.82	90.35
	201,000–400,000 cells/ml	3	23.09 (12.39)	6.07	58.23
	>400,000 cells/ml	1	7.35 (7.43)	0.71	46.68
Mastitis is one of reasons to cull cows		19	94.64 (5.36)	65.29	99.40
Average percentage of annual culling due to mastitis		17	3.52 (1.16)	1.09	5.96

^1^ CMT: California mastitis test,^2^ PCR: polymerase chain reaction, ^3^ SCM: subclinical mastitis, ^4^ CM: clinical mastitis, ^5^ IMM AMD: intramammary antimicrobial drugs, ^6^ AMD: antimicrobial drugs, ^7^ others: vitamins, minerals, local preparations, ^8^ others: electronic device attached to each cow.

**Table 5 vetsci-10-00629-t005:** Summary of responses on management practices around dry off period from a questionnaire on milk quality and mastitis management on Egyptian dairy farms (n = 20).

Parameter	Levels	N	Estimate (%)	95% CI	
				Lower	Upper
Dry off protocol	Gradual stop milking	17	84.43 (9.02)	55.69	95.90
	Sudden and gradual stop milking	3	15.57 (9.02)	4.10	44.31
Average length of dry period	<60 days	2	9.73 (6.92)	1.99	36.39
	60–65 days	14	70.54 (10.53)	44.99	87.51
	67–73 days	4	19.73 (9.24)	6.66	45.84
Internal or external teat sealant	Yes	2	9.37 (6.64)	1.93	35.18
Dry cow IMM AMD	Yes	20	100	.	.
Common dry cow IMM AMD ^1^	Cepravin DC	12	59.82 (11.53)	35.00	80.45
	Spectramast DC	8	40.45 (12.14)	18.92	66.41
	Orbenin DC	7	33.48 (10.50)	15.63	57.77
	Mummiforte DC	2	10.71 (6.92)	2.53	35.72
	Bovaclox DC	1	5.00 (5.00)	0.56	32.89
Dry off therapy protocol	Blanket dry cow therapy	20	100	.	.
Mastitis vaccination	Yes	9	45.18 (11.88)	22.96	69.50
Mastitis vaccine	Lysigin^®^	9	100	.	.
Dry off treatment reduce mastitis and increase milk production	Yes	19	95.00 (5.00)	67.11	99.44
	No	1	5.00 (5.00)	0.56	32.89
Mean reduction in mastitis case rate	(%)	9	35.75 (10.61)	11.27	60.22
Mean increase in milk production	(%)	6	9.83 (2.31)	3.88	15.78

^1^ IMM AMD: intramammary antimicrobial drugs.

**Table 6 vetsci-10-00629-t006:** Summary of responses to questions related dairy management practices from a questionnaire on milk quality and mastitis management practices on Egyptian dairies (n = 20).

Parameter	Levels	N	Estimate (%)	95% CI	
				Lower	Upper
Bedding materials used for lactating cows	Sand	13	64.64 (9.33)	43.49	81.29
	Soil	4	20.36 (9.33)	7.02	46.40
	Sand and soil	1	5.00 (5.00)	0.56	32.89
	Soil and rice hulls	2	10.00 (6.12)	2.56	31.98
Bedding materials used for dry cows	Sand	13	64.64 (9.33)	43.49	81.29
	Soil	3	15.36 (8.87)	4.09	43.53
	Sand and soil	2	10.00 (7.07)	2.06	37.01
	Soil and rice hulls	2	10.00 (6.12)	2.56	31.98
Bedding materials used for close-up cows	Sand	13	64.64 (9.33)	43.49	81.29
	Soil	2	10.36 (7.33)	2.12	38.11
	Sand and soil	2	10.00 (7.07)	2.06	37.01
	Soil and rice hulls	2	10.00 (6.12)	2.56	31.98
	Straw	1	5.00 (5.00)	0.56	32.89
Change the bedding materials during summer	Times/year	19	1.94 (0.59)	0.68	3.20
Change the bedding materials during winter	Times/year	19	6.23 (1.98)	2.06	10.41
What are the sources for replacement on your dairy?	Raise own heifers	20	100		
	Buy replacements (importation)	11	56.16 (11.38)	32.47	77.34
What are types of animals purchased for replacement?	Pregnant cows	3	14.37 (7.24)	4.60	36.89
	Pregnant heifers	8	41.79 (10.50)	22.32	64.19
Do you quarantine newly purchased animals?	Yes	9	82.67 (11.80)	43.24	96.76
	No	2	17.33 (11.80)	3.24	56.76
Do you culture milk from newly purchased animals?	Yes	1	7.79 (7.90)	0.72	49.50
	No	10	92.2 (7.90)	50.50	99.28
Do you keep records of newly purchased animal medication?	Yes	8	80.00 (12.65)	40.08	95.99
	No	2	20.00 (12.65)	4.01	59.92
Do you feed cows directly after milking?	Yes	20	100	.	.
Where you feed cows after milking?	In the barn	19	95.63 (4.37)	70.43	99.50
	On the way back from parlor	1	4.37 (4.37)	0.50	29.57
How many times per day do you feed the lactating cows?	3–5 times/day	12	60.18 (12.06)	34.21	81.45
	6–7 times/day	5	25.09 (10.22)	9.56	51.48
	10–12 times/day	3	14.73 (8.53)	3.93	42.18
How you track withdrawal periods for AMD ^1^ used to treat mastitis?	Computer system	9	43.57 (10.36)	24.01	65.36
	Paper records	3	15.71 (8.53)	4.54	42.22
	Computer system and paper records	8	40.71 (10.30)	21.74	62.93

^1^ AMD: antimicrobial drugs.

**Table 7 vetsci-10-00629-t007:** Summary of multiple factor analysis of survey responses showing five identified components extracted from 73 variables from the survey of the milk quality and mastitis management practices on Egyptian dairies (n = 20).

Identified Components	Variation Proportion (%)	Component Variables	Correlation
Herd Demographics 1	11.21	District	0.41
		How do you milk the fresh cow	0.40
		How often do you dry off cows?	0.45
Milking Practices 1	19.78	Do milkers wear gloves during milking?	0.48
		Do you make culture swabs from tanks and milk lines?	0.50
		Pre-dip method	0.71
		Pre-dip disinfectant base	0.62
Milking Practices 2	19.42	How many milking units in the parlor?	0.78
		How many workers in the milking parlor?	0.65
		Number of work shifts per worker	−0.79
		How many hours does each milker complete per shift?	0.75
		Time elapsed between pre-dip and put the milking machine on	0.65
		Milk storage temperature in the milk tank	−0.48
		Frequency of replacing the liners of the milking cups	0.47
Management Practices 1	22.69	What are bedding material used for close-up cows?	0.82
		What are bedding material used for lactating cows?	0.80
		What are bedding material used for dry cows?	0.80
Management Practices 2	14.79	How often do you change bedding material in winter?	0.69
		How often do you change bedding material in summer?	0.71
		How many times per day do you feed lactating cows?	0.65

**Table 8 vetsci-10-00629-t008:** Summary statistics of the two main clusters identified by the hierarchical cluster analysis of the multiple factor analysis outcomes of the milk quality and mastitis management practices on Egyptian dairies (n = 20).

Components	Component Variables	Levels	Cluster 1 (n = 14), % (SE)	Cluster 2 (n = 6) % (SE)
Herd Demographics 1	Farm district	Ismailia	28.57 (12.07)	0.00
		Fayoum	14.29 (9.35)	33.33 (19.25)
		Alexandria	42.86 (13.23)	16.67 (15.21)
		Delta	14.29 (9.35)	50.00 (20.41)
	How often do you dry off cows?	Weekly or every 2 weeks	35.71 (12.81)	33.33 (19.25)
		Monthly	35.71 (12.81)	0.00
		As needed	28.57 (12.07)	66.67 (19.25)
	How do you milk the fresh cows?	At the beginning of milking	42.86 (13.23)	50.00 (20.41)
		At the end of milking	28.57 (12.07)	16.67 (15.21)
		Mobile milking machine	7.14 (6.88)	16.67 (15.21)
		Separate parlor	21.43 (10.97)	16.67 (15.21)
Milking Practices 1	Do milkers wear gloves during milking?	No	28.57 (12.07)	100.00 (0.00)
		Yes	71.43 (12.07)	0.00
	Do you make culture swabs from tanks and milk lines?	No	57.14 (13.23)	66.67 (19.25)
		Yes	42.86 (13.23)	33.33 (19.25)
	Pre-dip method	None	0.00	16.67 (15.21)
		Cup	92.86 (6.88)	83.33 (15.21)
		Spray	7.14 (6.88)	0.00
	Pre-dip disinfectant base	Iodine	85.71 (9.35)	83.33 (15.21)
		Iodine and chlorhexidine	7.14 (6.88)	0.00
		Lactic acid	7.14 (6.88)	0.00
		None	0.00	16.67 (15.21)
Milking Practices 2	How many milking units in the parlor?	0–20 units	42.86 (13.23)	83.33 (15.21)
		21–48 units	57.14 (13.23)	16.67 (15.21)
	How many workers in the milking parlor?	2–10 workers	78.57 (10.97)	83.33 (15.21)
		10–35 workers	21.43 (10.97)	16.67 (15.21)
	How many work shifts per worker?	One shift	28.57 (12.07)	33.33 (19.25)
		Two shifts	42.86 (13.23)	33.33 (19.25)
		Three shifts	28.57 (12.07)	33.33 (19.25)
	How many hours does each milker complete per shift?	2–4 h	64.29 (12.81)	83.33 (15.21)
		5–8 h	35.71 (12.81)	16.67 (15.21)
	Average time elapsed between pre-dip and application of milking machine	One minute or less	71.43 (12.07)	66.67 (19.25)
		More than one minute	28.57 (12.07)	33.33 (19.25)
	Parlor maintenance: frequency of replacing liners of milking cups	Every 2000–3000 milking times	64.29 (12.81)	33.33 (19.25)
		Every 3200–6000 milking times	35.71 (12.81)	66.67 (19.25)
	Milk storage temperature in the milk tank	Mean (SE) °C	3.07 (0.115)	2.50 (0.223)
Management Practices 1	What are bedding material used for close-up cows?	Sand	92.86 (6.88)	0.00
		Sand and soil	0.00	33.33 (19.25)
		Soil, agriculture soil, clay	0.00	33.33 (19.25)
		Soil and rice hulls	7.14 (6.88)	16.67 (15.21)
		Straw	0.00	16.67 (15.21)
	What are bedding material used for lactating cows?	Sand	92.86 (6.88)	0.00
		Sand and soil	0.00	16.67 (15.21
		Soil, agriculture soil, clay	0.00	66.67 (19.25)
		Soil and rice hulls	7.14 (6.88)	16.67 (15.21)
	What are bedding material used for dry cows?	Sand	92.86 (6.88)	0.00
		Sand and soil	0.00	33.33 (19.25)
		Soil, agriculture soil, clay	0.00	50.00 (20.41)
		Soil and rice hulls	7.14 (6.88)	16.67 (15.21)
Management Practices 2	How often do you change bedding material in winter?	Less than 12 times	100.00 (0.00	50.00 (20.41)
		More than 12 times	0.00	50.00 (20.41)
	How often do you change bedding material in Summer?	Less than 12 times	100.00 (0.00)	83.33 (15.21)
		More than 12 times	0.00	16.67 (15.21
	How many times per day do you feed lactating cows?	3–6 times	71.43 (12.07)	83.33 (15.21)
		More than 6 times	28.57 (12.07)	16.67 (15.21)

## Data Availability

Individual data collected are confidential to prevent individual identification of a farm or business; as such, raw data from this study are not able to be shared. Summary data are available upon reasonable request.

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
