# Peer review of "Management Practices of Bovine Mastitis and Milk Quality on Egyptian Dairies"

_vetsci, 2023, doi:10.3390/vetsci10100629_

Round 1
Reviewer 1 Report
in attachment

Author Response
Reviewer, Title: Management practices of mastitis and milk quality on Egyptian dairies
This manuscript looks like very well and it is good written and very clear in all parts. Manuscript is so long and should be shorter.
Manuscript is based on the survey on Egyptian dairies manangments for mastitis and milk quality. The survey consists four directions: herd demographics; parlor management and milking practices; mastitis management practices; dry off management practices and in the end the multiple factor analysis (MFA).
Dairy herds devided in to two hierarchica clusters but in discussion and in the conclusions did not describe differences between them.
Authors: Thanks for alerting us to this, we have now summarized our cluster analysis findings in the conclusions.
Reviewer: In my opinion manuscript based on the survey should be submit in journals with lowest impact factor or national journals. But, it is first article in Egypt state about management practices of mastitis and milk quality on egyptian dairies, I will give a positive opinion.
Authors: Thanks, your approval for our findings’ uniqueness is appreciated.
Reviewer 2 Report
In this manuscript, the management of Egyptian cows, mastitis management and milk quality management were investigated in detail, and the relevant preventive measures were given. The results of this paper have certain guiding significance to the production practice. Here are the problems:1. In the discussion section, the management of mastitis is only discussed, and the management strategy of milk quality is not enough.
2. In the result section, the secondary title was only 3.1, so why make it a tertiary title?
3. Less than 1/3 of the references were cited in the last five years, and more references were cited ten years ago.
4. "References" is repeated twice. Is it overwritten? Why does the DOI number in document [42] have to be added to another line?
Moderate editing of English language required.
Author Response
In this manuscript, the management of Egyptian cows, mastitis management and milk quality management were investigated in detail, and the relevant preventive measures were given. The results of this paper have certain guiding significance to the production practice. Here are the problems:
Reviewer: 1. In the discussion section, the management of mastitis is only discussed, and the management strategy of milk quality is not enough.
Authors: We now identify the milk quality factors* and reference them in discussion lines 410 to 452 of the word documents section 3.1.2 “Parlor management and milking practices”
(pre- and post-milking udder hygiene including udder wash, pre-dip, stripping of the foremilk, wiping of teats, time between pre-dip and application of milking machine, and post dip)
Reviewer: 2. In the result section, the secondary title was only 3.1, so why make it a tertiary title?
Authors: Thanks for catching the sectioning of the manuscript, because we submitted a word document without numbering sections we believe this will need to be corrected by the editorial staff.
Reviewer: 3. Less than 1/3 of the references were cited in the last five years, and more references were cited ten years ago.
Authors: We revisited and modified the references extensively keeping in mind use of most recent and replacing unless necessary older references.
Reviewer: 4. "References" is repeated twice. Is it overwritten? Why does the DOI number in document [42] have to be added to another line?
Authors: Corrected, thanks.
Reviewer 3 Report
In the manuscript ‘Management practices of mastitis and milk quality on Egyptian dairies’ is described the herd demographics and different management practices related to mastitis control and prevention on Egyptian cow dairy herds. This is an interesting, thoroughly analysed, and comprehensive study. However, is the milk in Egyptian dairies only of bovine origin or is it also from sheep/goats? This needs to be clarified in the manuscript and in the title, too.
Line 113: correct ‘manger’ to ‘manager’
Another parameter that could be examined in 3.1.2. and Table 3 would be the milker's training and education level, which greatly determines the observance of hygiene rules and the proper milking procedure. I understand that since this factor has not already been included in the research, it is difficult for the authors to assess it retrospectively.
Author Response
Reviewer: In the manuscript ‘Management practices of mastitis and milk quality on Egyptian dairies’ is described the herd demographics and different management practices related to mastitis control and prevention on Egyptian cow dairy herds. This is an interesting, thoroughly analysed, and comprehensive study. However, is the milk in Egyptian dairies only of bovine origin or is it also from sheep/goats? This needs to be clarified in the manuscript and in the title, too.
Authors: Thanks, we have edited the title and confirmed that the abstract and the text now identify the species of interest as bovine or cow at first instance.
Reviewer: Line 113: correct ‘manger’ to ‘manager’
Authors: Corrected, Thanks.
Reviewer: Another parameter that could be examined in 3.1.2. and Table 3 would be the milker's training and education level, which greatly determines the observance of hygiene rules and the proper milking procedure. I understand that since this factor has not already been included in the research, it is difficult for the authors to assess it retrospectively.
Authors: thanks for your suggestion, we will consider that in future studies.